# The qualitative assessment of optical coherence tomography and the central retinal sensitivity in patients with retinitis pigmentosa

**Aiko Hara, Mitsuru Nakazawa* , Masaaki Saito, Yukihiko Suzuki**

Department of Ophthalmology, Hirosaki University Graduate School of Medicine, Hirosaki, Japan

* mitsuru@hirosaki-u.ac.jp

**Data Availability Statement:** All relevant data are within the paper and its Supporting Information files.

## Abstract

### Purpose

To analyze the relationships between qualitative and quantitative parameters of spectral-domain optical coherence tomography (SD-OCT) and the central retinal sensitivity in patients with retinitis pigmentosa (RP).

### Materials and methods

Ninety-three eyes of 93 patients were finally enrolled, with a median age (quartile) of 58 (24.5) years. We assessed the patients using SD-OCT and the 10–2 program of a Humphry Field Analyzer (HFA). As a qualitative parameter, two graders independently classified the patients' SD-OCT images into five severity grades (grades 1–5) based on the severity of damage to the photoreceptor inner and outer segments (IS/OS) layer. As quantitative parameters, we measured the IS-ellipsoid zone (IS-EZ) width, IS/OS thickness, outer nuclear layer (ONL) thickness, central macular thickness (CMT, 1 and 3 mm) and macular cube (6 × 6 mm) volume and thickness. The central retinal sensitivity was defined by the best-corrected visual acuity (BCVA; logMAR), average sensitivities of the central 4 (foveal sensitivity [FS]) and 12 (macular sensitivity [MS]) points of the HFA 10–2 program and the mean deviation (MD) of the 10–2 program. Spearman's correlation was used to assess the association between both qualitative and quantitative parameters and variables of the central retinal sensitivity. In addition, we performed a multiple regression analysis using these parameters to identify the parameters most strongly influencing the central retinal sensitivity.

### Results

The IS/OS severity grade was significantly correlated with the BCVA ($\rho = 0.741$, $P < 0.001$), FS ($\rho = -0.844$, $P < 0.001$), MS ($\rho = -0.820$, $P < 0.001$) and MD ($\rho = -0.681$, $P < 0.001$) and showed stronger correlations to them than any other quantitative parameters including the IS-EZ width, IS/OS thickness, ONL thickness, CMTs and macular cube volume/thickness. Furthermore, a step-wise multiple regression analysis indicated that the IS/OS severity grade was more strongly associated with the BCVA ($\beta = 0.659$, $P < 0.001$), FS ($\beta = -0.820$,

**Funding:** This study was supported, in part, by the Grant-in-Aids for Scientific Research 19K09926 (MN) and 16K11257 (MS) from the Japan Society for the Promotion of Science (Tokyo, Japan: https://www.jsps.go.jp/) and Research grants from Alcon (MN; Tokyo Japan), Novartis (MN; Tokyo, Japan), Santen (MN; Osaka, Japan), AMO (MN; Tokyo, Japan), HOYA (MN; Tokyo, Japan) and Pfizer (MN; Tokyo, Japan). The funders had no role in study design, data collection and analysis, decision to publish, or preparation of the manuscript.

**Competing interests:** Mitsuru Nakazawa received research funds from Alcon (Tokyo, Japan), Novartis (Tokyo, Japan), Santen (Osaka, Japan), AMO (Tokyo, Japan), HOYA (Tokyo, Japan) and Pfizer (Tokyo, Japan). The funders had no role in study design, data collection and analysis, decision to publish or preparation of the manuscript. This does not alter our adherence to PLOS ONE policies on sharing data and materials.

$P < 0.001$), MS ($\beta = -0.820$, $P < 0.001$) and MD ($\beta = -0.674$, $P < 0.001$) than any other quantitative parameters. The intraclass correlation coefficient between two graders indicated substantial correlation ($\kappa = 0.70$).

## Discussion

The qualitative grading of OCT based on the severity of the IS/OS layer was simple and strongly correlated with the central retinal sensitivity in patients with RP. It may be useful to assess the central visual function in patients with RP, although there is some variation in severity within the same severity grade.

## Introduction

Retinitis pigmentosa (RP) is a clinical entity including genetically and phenotypically heterogenous groups of slowly progressive hereditary photoreceptor degeneration. Most patients with RP initially show rod-predominant degeneration and subsequently followed by cone degeneration. Clinically, they initially complain of photophobia and visual disturbance under dim circumstance followed by concentric constriction of the visual field due to progressive rod-predominant degeneration. As the disease progresses, decreased visual acuity and color blindness become obvious due to consecutive cone degeneration. Because the central visual field is important for daily life activity in most people, patients with RP tend to experience difficulty in their daily life activity once their central visual field becomes impaired. Therefore, assessing a patient's central visual function is important for developing a treatment program, evaluating the effects of treatment and explaining the disease status to patients.

Static visual field tests like the Humphry Field Analyzer® (HFA; Carl Zeiss Meditec, Dublin, CA, USA) or Octopus Visual Field Analyzer® (Haag-Streit Diagnostics, Mason, OH, USA) are probably the most frequently used to quantitatively evaluate the central visual function in patients with RP. Although static visual field tests theoretically provide exact visual sensitivities at preprogrammed visual field points, they require extensive concentration from patients and are greatly affected by the patient's general condition and levels of cooperation, as visual field tests have a psychophysical nature.

To overcome these potential problems, relationships between the structure and function have been extensively investigated in order to utilize the objective and quantitative parameters obtained from spectral-domain optical coherence tomography (SD-OCT) to assess the central retinal sensitivity of patients with RP [1–24]. Previously, among several quantitative parameters, the width of the inner segment ellipsoid zone (IS-EZ) [1, 3, 7, 8, 10, 13, 14, 18, 19, 21, 24] and thickness of the outer retinal layer [2, 4, 9, 12, 15, 20, 23] have been shown to be significantly correlated with the central visual sensitivity. However, in these quantitative analyses, the subjects should be restricted to patients with visual acuity better than a certain level or measurable structural changes for analyzing the correlation between structural parameters and the visual function [1, 9, 10, 12, 13, 15, 18, 20, 21]. In addition, while Sandberg et al. and Aizawa et al. have previously reported the correlation between the qualitative changes in time-domain OCT (TD-OCT) findings and central visual acuity in patients with RP [1, 7], the resolution of TD-OCT was limited compared with that of SD-OCT. Although these clinical studies have provided important information regarding the relationships between the structure and function, the precise structural changes obtained from SD-OCT have not been fully clarified, as human retinal tissue cannot be excised for a histopathological examination.

To overcome these points, animal models harboring homologous genetic mutations to human RP patients may provide some clues for understanding the relationship between SD-OCT images and their histopathological backgrounds caused by photoreceptor degeneration [25]. Based on these speculations, the findings from previously performed animal experiments have enabled us to correlate the pathological findings of photoreceptor degeneration with the findings from SD-OCT [25–30]. Common findings in model animals for human RP associated with various kinds of gene mutations include diffuse hyperreflective changes in the photoreceptor inner and outer segments (IS/OS) layer and thinning of the outer nuclear layer (ONL) [25]. As degeneration of the photoreceptor progresses, although the IS-EZ and inter-digitation zone (IZ) are initially identified as two independent discrete lines, they gradually broaden and finally merge into a single diffuse hyperreflective zone in the IS/OS layer [26–28]. The diffuse hyperreflective changes in the IS/OS layer are considered to be the same phenomenon as merging of the IS-EZ and IZ [31]. We considered these characteristic changes in the IS/OS layer on SD-OCT observed in animal models of RP to be applicable for classifying the severity of morphologic changes in patients with RP in greater detail than those previously reported [1, 7]. Therefore, in the present study, we created a qualitative grading system according to severity of the IS/OS layer in SD-OCT images seen in patients with RP.

We analyzed the correlation between the qualitative severity grade and the variables of the central retinal sensitivity calculated from the visual acuity and results of an HFA. We additionally compared the qualitative severity grade with other quantitative parameters to examine the suitability of using the qualitative severity grade to evaluate the central retinal sensitivity in patients with RP.

## Materials and methods

This study was a retrospective observational case series. The study protocols adhered to the tenets of the Declaration of Helsinki and were approved by the Institutional Review Board at the Hirosaki University (#2017–1079). Written informed consent was obtained from all of the final participants in the study.

### Subjects

We reviewed the medical records of 130 patients with RP who were regularly examined at Hirosaki University Hospital between April 2017 and October 2019. The diagnosis of RP was based on the presence of night blindness, typical fundus features of RP (including mottled appearance of the mid-peripheral and/or peripheral fundi bilaterally with or without bone-spicule like pigmentary deposits), concentric visual field constriction and reduced or extinguished amplitudes of dark-adapted full-field standard electroretinography [32]. Of these 130 patients, 93 were considered eligible because they had undergone SD-OCT (Cirrus™ HD-OCT, Model 5000; Carl Zeiss Meditec) and been examined by the 10–2 SITA Standard Program of the HFA (HFA II 7501, Carl Zeiss Meditec) on the same day or during the same period at no more than 3 months apart.

We excluded eyes with epiretinal membrane, macular hole or retinoschisis, cystoid macular edema, severe opacity in the ocular media or ocular diseases other than RP. We also excluded patients whose HFA data showed fixation loss scores of $\geq$ 20% and/or a false-negative error $\geq$ 30%. For further analyses, we used 93 eyes of 93 patients. The median age (quartile) was 58 (24.5) years old, and the oldest patient was 85 years old and the youngest 12 years old. Forty-eight patients were female (female/male = 48/45). Based on the family history, the patients were classified as having autosomal dominant RP (n = 7), autosomal recessive RP (n = 5), X-linked RP (n = 1) and RP simplex (n = 80). Although the data of the right eye were

basically used for analyses, if the right eye was not suitable for our analyses due to medical reasons mentioned among the exclusion criteria, then the data of the left eye were used.

## Central retinal sensitivity

Variables of central retinal sensitivity consisted of the best-corrected visual acuity (BCVA), the averages of the visual sensitivities (dB) of the central 4 points (foveal sensitivity [FS]) [22] and the central 12 points (macular sensitivity [MS]) [22] of the 10–2 Program of the HFA and mean deviation (MD), as described in Fig 1. The BCVA was measured using a Landolt decimal visual acuity chart (TCU-500; Fuji Kogaku, Tokyo, Japan or VC-60; Takagi Seiko, Nakano,

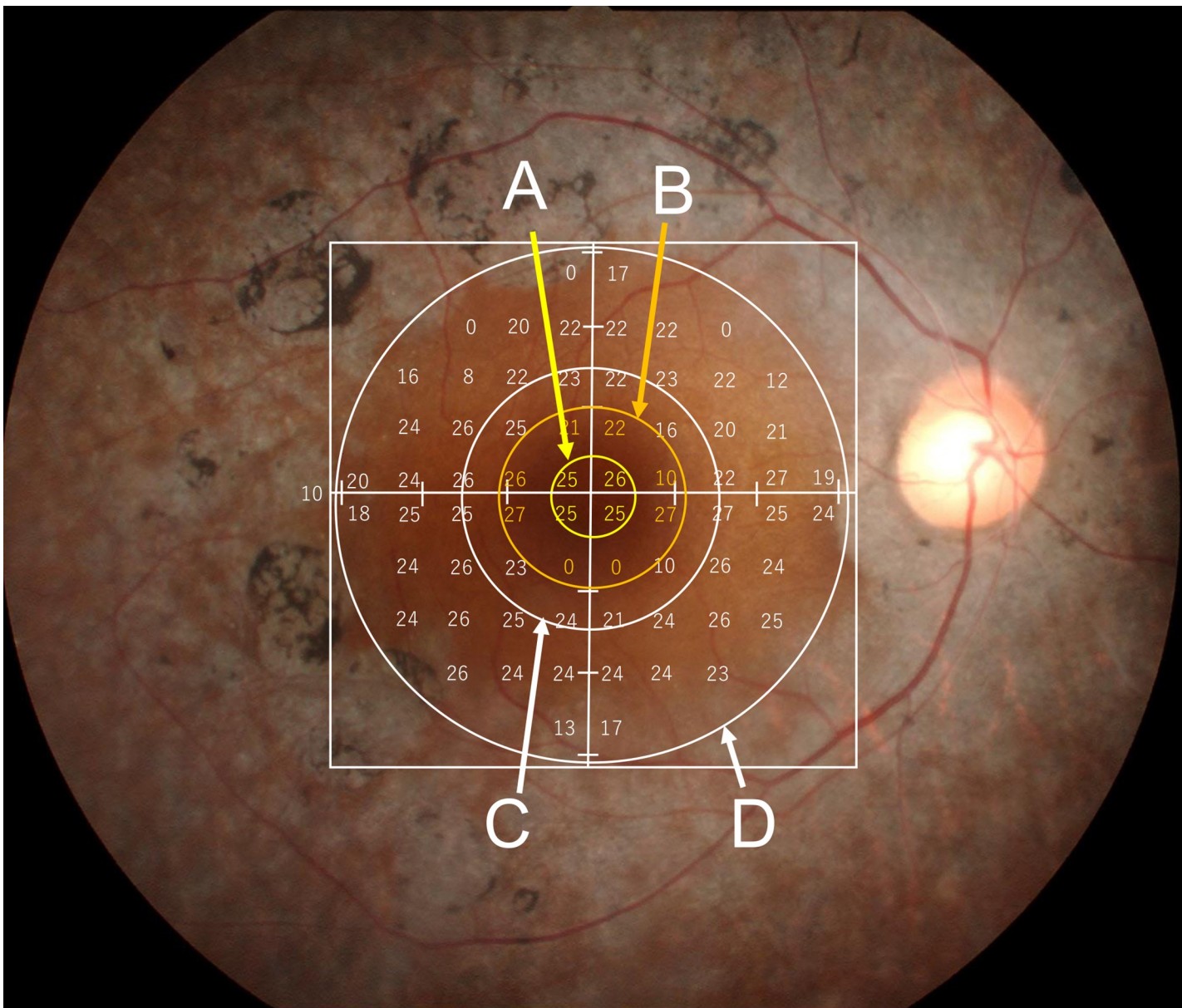

**Fig 1.** Fundus picture of a patient with RP overlaid by the same patient's 10–2 plots of HFA and circles with radii of 1.0 mm (A), 2.2 mm (B), 3.0 mm (C) and 6.0 mm (D). An outermost squire is the macular cube (6 × 6 mm) of SD-OCT.

Japan) at 5 m or single Landolt cards (HP-1258; Handaya, Tokyo, Japan). The values of BCVA were converted to the logarithm of the minimum angle of resolution (logMAR). If the BCVA was Counting Fingers at 30 cm or Hand Motion level, the logMAR was set at 2.5 or 3.0, respectively.

Fig 1 shows a fundus image overlaid with an example of the 10–2 program visual field test location. The FS and MS correspond to the retinal sensitivity within a circle with 1.0-mm radius (circle A) and within a circle of 2.2-mm radius (circle B), respectively. The MD value corresponds to the averaged retinal sensitivity within a circle with 3.0-radius (circle D).

### Qualitative severity grading of SD-OCT findings

Based on the results from previous animal experiments regarding the relationship between the histologic and ultrastructural findings of photoreceptor degeneration and corresponding SD-OCT features [25–30], we classified patients' horizontal and vertical sections of SD-OCT images into five severity grades mainly focused on the structure of the IS/OS layer. The definition of these five grades were as follows: grade 1, almost normal foveal structure or a sharply defined IS-EZ and IZ at the foveal center; grade 2, partial hyperreflective change in the IS/OS layer of the foveal region or a broadened IZ and/or IS-EZ without merging; grade 3, diffuse hyperreflective changes in the IS/OS layer or broad IS-EZ changes in the foveal region or merging of the broadened IS-EZ and IZ; grade 4, remnants of the IS/OS-like structure on the retinal pigment epithelium (RPE); grade 5, absent IS/OS-like structure.

The typical examples of SD-OCT figures are shown in Fig 2. Two experienced masked graders (AH and MN) independently evaluated patients' SD-OCT figures according to the criteria described above. If there were discrepancies between the graders' evaluation, the two graders discussed their opinions to reach a final common grade. For statistical analyses, we regarded the grade numbers (1–5) as semiquantitative severity scales for further calculations (IS/OS severity grade).

### Quantitative parameters of SD-OCT findings

As quantitative parameters of patients' SD-OCT images, we manually measured the width of the IS-EZ (μm), the thickness of the IS/OS layer (μm) at the central fovea and the thickness of

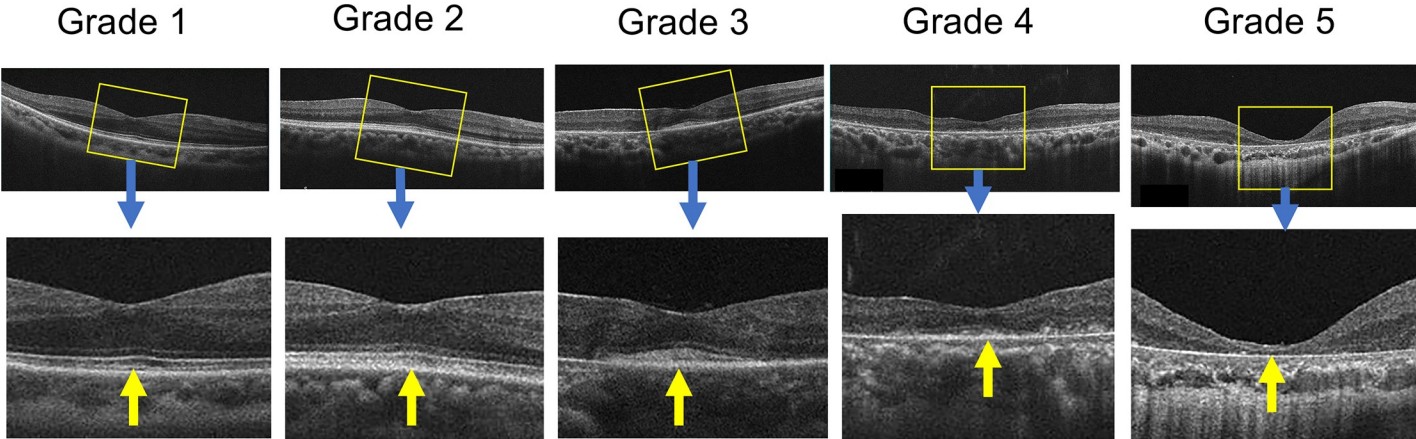

**Fig 2. Sample SD-OCT pictures indicating each severity grade.** Definition of grades: grade 1, almost a normal foveal structure or sharply defined IS-EZ and IZ at the foveal center (yellow arrow); grade 2, partial hyperreflective changes in the IS/OS layer of the foveal region or a broadened IZ (yellow arrow) and/or IS-EZ without merging; grade 3, diffuse hyperreflective changes in the IS/OS layer or broad IS-EZ changes in the foveal region or merging of the broadened IS-EZ and IZ (yellow arrow); grade 4, remnants of the IS/OS-like structure on the retinal pigment epithelium (yellow arrow); grade 5, absent IS/OS-like structure (yellow arrow).

the ONL (μm) at the center of the fovea using horizontal sections of patients' SD-OCT images (Fig 3) and ImageJ® software program (U.S. National Institute of Health, Bethesda, MA, USA). The IS-EZ width was defined as the horizontal distance between the temporal and nasal boarders of the IS-EZ where the IS/OS contour disappeared. If the IS-EZ was longer than 6,000 μm, the value was fixed at 6,000 μm. The IS/OS layer thickness was defined as the distance between the external limiting membrane (ELM) and the superficial boarder of RPE, as the photoreceptor inner and outer segments are normally located in the space between ELM and RPE. The ONL thickness was defined as the distance between the deep boarder of the outer plexiform layer and ELM. If there were no obvious findings of the IS-EZ, IS/OS layer or ONL to be measured on an SD-OCT image, we assigned each parameter a value of 0.

In addition, we adopted the averages of the central retinal thickness (CMT, μm) in the 1-mm (circle A, Fig 1) and 3-mm (circle C, Fig 1) circles (thickness from the inner limiting membrane [ILM] to RPE) in patients' SD-OCT data. We also used the averages of the retinal volume (mm²) and thickness (μm) of the macular cube (6 × 6 mm, from ILM to RPE) in patients' SD-OCT data (Fig 1).

## Statistical analyses

Statistical analyses were performed using the SPSS version 26 software (Statistical Package for the Social Science, IBM Corp, Armonk, NY, USA). The Shapiro-Wilk test was used to assess

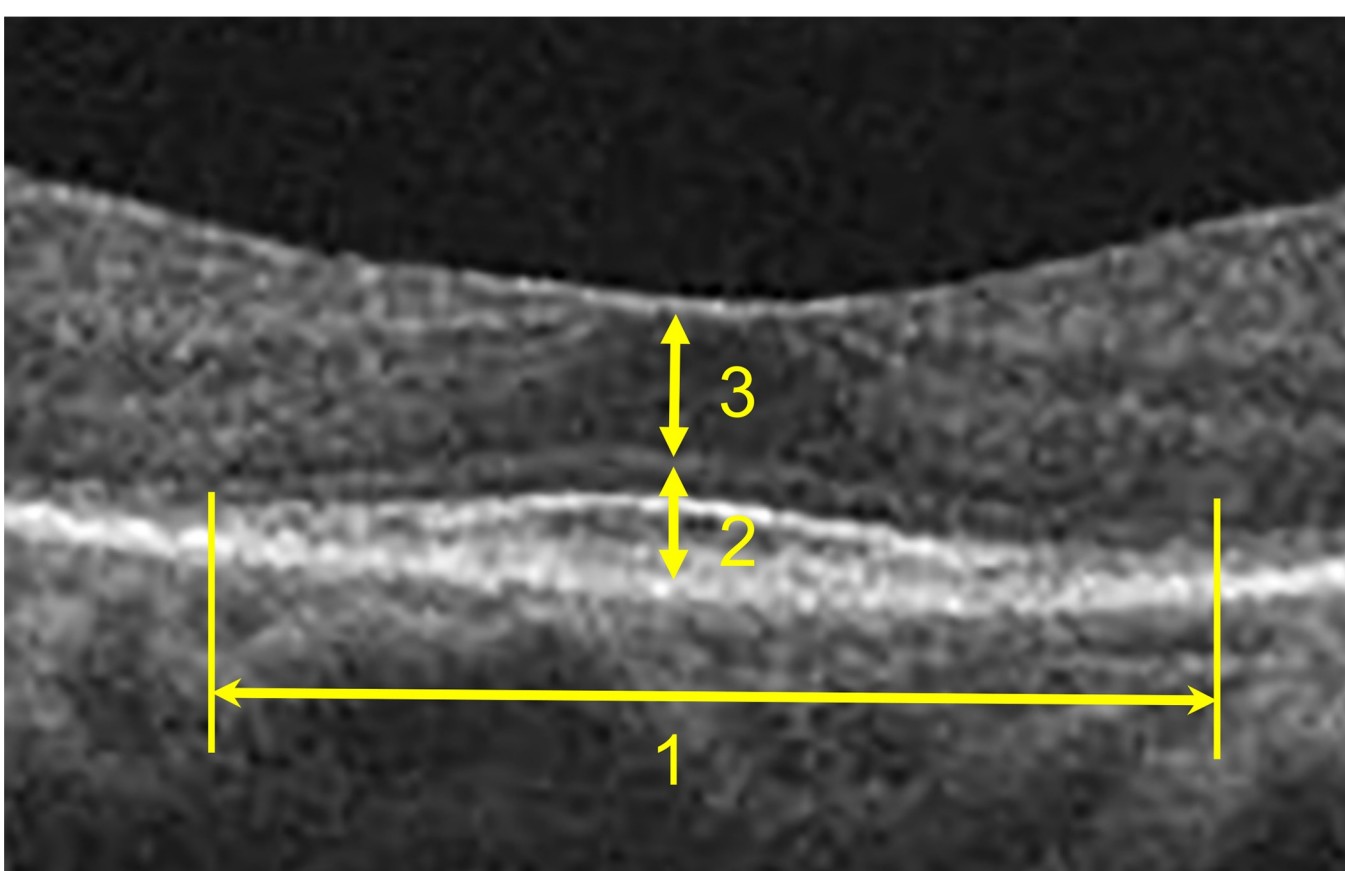

**Fig 3. Diagrams of quantitative SD-OCT parameters.** 1, width of inner segment ellipsoid zone (IS-EZ); 2, thickness of photoreceptor inner and outer segments (IS/OS) layer; 3, thickness of the outer nuclear layer (ONL).

the normality of the distribution of all acquired data sets. The agreement of severity grade between two graders was assessed by intraclass correlation coefficients (ICC, class 2) using the κ coefficient, and κ > 0.61 was considered as the threshold for substantial agreement. Values of variables reflecting the central retinal sensitivity in each qualitative severity grade group were compared by a one-way analysis of variance using the Kruskal-Wallis test, followed by Bonferroni's test as a *post hoc* multiple comparison analysis. A Spearman's correlation rank test was used to examine the association between the qualitative severity grade score or quantitative parameters of SD-OCT images and variables reflecting the central retinal sensitivity, and $0.40 < |\rho| \leq 0.7$ was prespecified as "well correlated" while $0.7 < |\rho| < 1.0$ was considered as "highly correlated". While, strictly speaking, the qualitative severity grade score is not an interval scale but an ordinal scale, we considered that the qualitative severity grade score to be treatable as a semi-interval scale followed by assessment by the Spearman's correlation rank test.

In addition, a step-wise multiple regression analysis was performed to identify which qualitative or quantitative parameters of SD-OCT images most strongly influenced each of the variables of the central retinal sensitivity. In this analysis, we also considered the qualitative severity grade score as a semi-interval scale and applied a multiple regression analysis as well. $P < 0.05$ was considered statistically significant.

# Results

## Demographic features of the study eyes

The demographic data of the study eyes are summarized in Table 1. Because most of the data did not show a normal distribution, the data are presented as median (quartile). Raw data of the quantitative parameters of the patients' SD-OCT findings and the variables of the central retinal sensitivities are presented in S1 and S2 Tables. For the qualitative IS/OS severity grade,

**Table 1. Characteristics of the 93 patients with RP.**

| | |
|---|---|
| Patients, n | 93 |
| Sex, female (%) | 48 (51.6) |
| Age | 58 (24.5) |
| Central Retinal Sensitivity | |
| BCVA, logMAR | 0.22 (0.61) |
| FS, dB | 26.00 (21.00) |
| MS, dB | 20.75 (23.25) |
| MD, dB | -22.84 (17.00) |
| Parameters of SD-OCT | |
| IS/OS severity grade | 2.00 (3.00) |
| IS-EZ width, μm | 2065 (3332) |
| IS/OS thickness, μm | 52.50 (41.25) |
| ONL thickness, μm | 85.00 (70.00) |
| CMT in 1mm, μm | 235.00 (90.00) |
| CMT in 3mm, μm | 276.44 (66.67) |
| Cube volume, mm2 | 8.90 (1.90) |
| Cube thickness, μm | 248.00 (51.00) |

Values are given median (quartile) except patient number and age.

Abbreviations: BCVA, best corrected visual acuity; FS, foveal sensitivity; MS, macular sensitivity; MD, mean deviation of HFA 10–2; IS/OS, inner segment/outer segment; IS-EZ, inner segment ellipsoid zone; ONL, outer nuclearlayer; CMT, central macular thickness.

patients were distributed as follows: grade 1, n = 13; grade 2, n = 33; grade 3, n = 16; grade 4, n = 20; and grade 5, n = 11 (S3 Table).

### Correlation between structure and function

The median values of the BCVA, FS, MS or MD in each qualitative IS/OS severity group were significantly different ($P < 0.001$, Kruskal-Wallis test). The relationships using box plots between values reflecting the central retinal function and the qualitative IS/OS severity grades are presented in Fig 4. Regarding the BCVA, the median value in grades 3 was significantly better than that in grade 4 ($P < 0.001$, Bonferroni's test, Fig 4). Regarding the FS and MS, the median values in grades 2 and 3 were significantly greater than those in grades 3 and 4, respectively ($P = 0.003$, FS, grade 2 vs. grade 3; $P < 0.001$, FS, grade 3 vs. grade 4, MS, grade 2 vs. grade 3, grade 3 vs. grade 4, Bonferroni's test, Fig 4). Regarding the MD, the median values in grades 1 and 2 were significantly greater than those in grades 2 and 3, respectively ($P = 0.004$, grade 1 vs. grade 2; $P = 0.032$, grade 2 vs. grade 3, Bonferroni's test, Fig 4). Given the results shown in Fig 4 apparently reflecting the central retinal function were correlated with the qualitative IS/OS severity grade, we considered it reasonable that the qualitative IS/OS severity grade was considered as a semi-quantitative value.

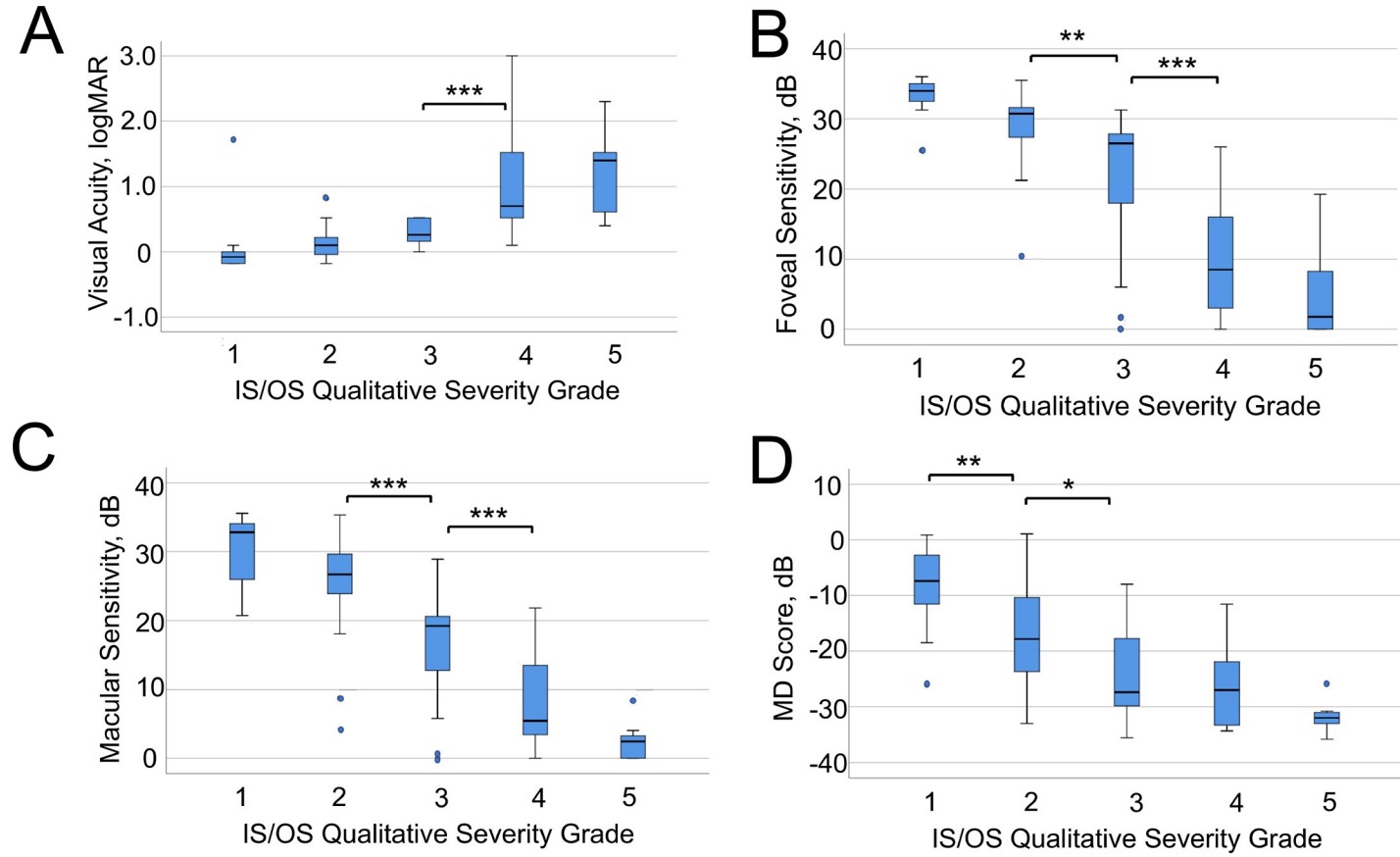

**Fig 4. Box plots showing the distributions of values of variables concerning the central visual function in each grade of the qualitative IS/OS severity grade.** A, BCVA, B, foveal sensitivity (FS), C, macular sensitivity (MS) and D, mean deviation (MD) of the 10–2 program of the HFA. The median value of each group is shown by the transverse line, and the box indicates the quartile range. Statistical significance; *, $P < 0.05$; **. $P < 0.01$; ***, $P < 0.001$.

The results of the Spearman's correlation analyses for the qualitative and quantitative parameters of SD-OCT and variables of central retinal sensitivity are shown in Table 2. The qualitative severity grade score (IS/OS severity grade) was significantly highly ($|\rho| > 0.7$) correlated with the BCVA ($\rho = 0.741$, $P < 0.001$), FS ($\rho = ^-0.844$, $P < 0.001$) and MS ($\rho = -0.820$, $P < 0.001$). In addition, the IS-EZ width was significantly highly correlated with the FS ($\rho = 0.742$, $P < 0.001$) and MS ($\rho = 0.765$, $P < 0.001$), the IS/OS thickness was significantly highly correlated with the BCVA ($\rho = -0.734$, $P < 0.001$), and the ONL thickness was significantly highly correlated with FS ($\rho = 0.731$, $P < 0.001$). Overall, the IS/OS severity grade, IS-EZ width, IS/OS thickness and ONL thickness showed "better than well" correlation with the BCVA, FS, MS and MD (Table 2). Although CMTs in 1- and 3-mm circles were well correlated ($0.40 < |\rho| \leq 0.7$) with most of the variables of the central retinal sensitivity, the macular cube volume and thickness did not show meaningful correlations with any of these variables (Table 2). Plot diagrams showing the correlations between the qualitative IS/OS severity grade and the variables of central retinal sensitivity are presented in Fig 5, and both the qualitative and quantitative parameters and the variables of central retinal sensitivity are shown in S1 and S2 Figs.

## Multiple regression analysis

We performed step-wise multiple regression analyses while taking all qualitative and quantitative parameters into account to identify factors affecting the variables of central retinal sensitivity. The results are summarized in Table 3. In model 1, only the IS-OS severity grade was selected as the most strongly influential parameter for each of the four variables. In model 2, in addition to the IS/OS severity grade, CMT in 1 mm central circle (circle A in Fig 1) and the IS-EZ width were chosen as explanatory parameters for the BCVA and MD, respectively. In Model 3, in addition to the parameters selected in Model 2, the ONL thickness was adopted as an explanatory factor for the BCVA. For FS and MS, no other parameters than the IS/OS severity grade were adopted as explanatory parameters.

## Agreement of grading between graders

The ICC analysis between 2 independent graders for grading qualitative severity of the IS/OS layer in SD-OCT pictures revealed a κ of 0.70 (match rate = 77.5%), indicating substantial agreement between the graders (S3 Table).

**Table 2. Correlations between parameters of SD-OCT and central retinal sensitivity.**

|  | BCVA | | FS | | MS | | MD | |
|---|---|---|---|---|---|---|---|---|
|  | ρ | *P*-value | ρ | *P*-value | ρ | *P*-value | ρ | *P*-value |
| IS/OS severity grade | 0.741 | <0.001 | −0.844 | <0.001 | −0.820 | <0.001 | −0.681 | <0.001 |
| IS-EZ width | −0.665 | <0.001 | 0.742 | <0.001 | 0.765 | <0.001 | 0.646 | <0.001 |
| IS/OS thickness | −0.734 | <0.001 | 0.691 | <0.001 | 0.657 | <0.001 | 0.539 | <0.001 |
| ONL thickness | −0.668 | <0.001 | 0.731 | <0.001 | 0.666 | <0.001 | 0.534 | <0.001 |
| CMT in 1mm | −0.493 | <0.001 | 0.586 | <0.001 | 0.584 | <0.001 | 0.407 | <0.001 |
| CMT in 3mm | −0.337 | 0.001 | 0.475 | <0.001 | 0.472 | <0.001 | 0.342 | 0.001 |
| Cube volume | −0.139 | 0.183 | 0.207 | 0.047 | 0.235 | 0.023 | 0.203 | 0.052 |
| Cube thickness | −0.140 | 0.182 | 0.209 | 0.044 | 0.236 | 0.022 | 0.205 | 0.049 |

ρ, correlation coefficient of a Spearman's correlation analysis; P-value, two-tailed P values < 0.05 were considered statistically significant.

Abbreviations: BCVA, best-corrected visual acuity; FS, foveal sensitivity; MS, macular sensitivity; MD, mean deviation of HFA 10–2.

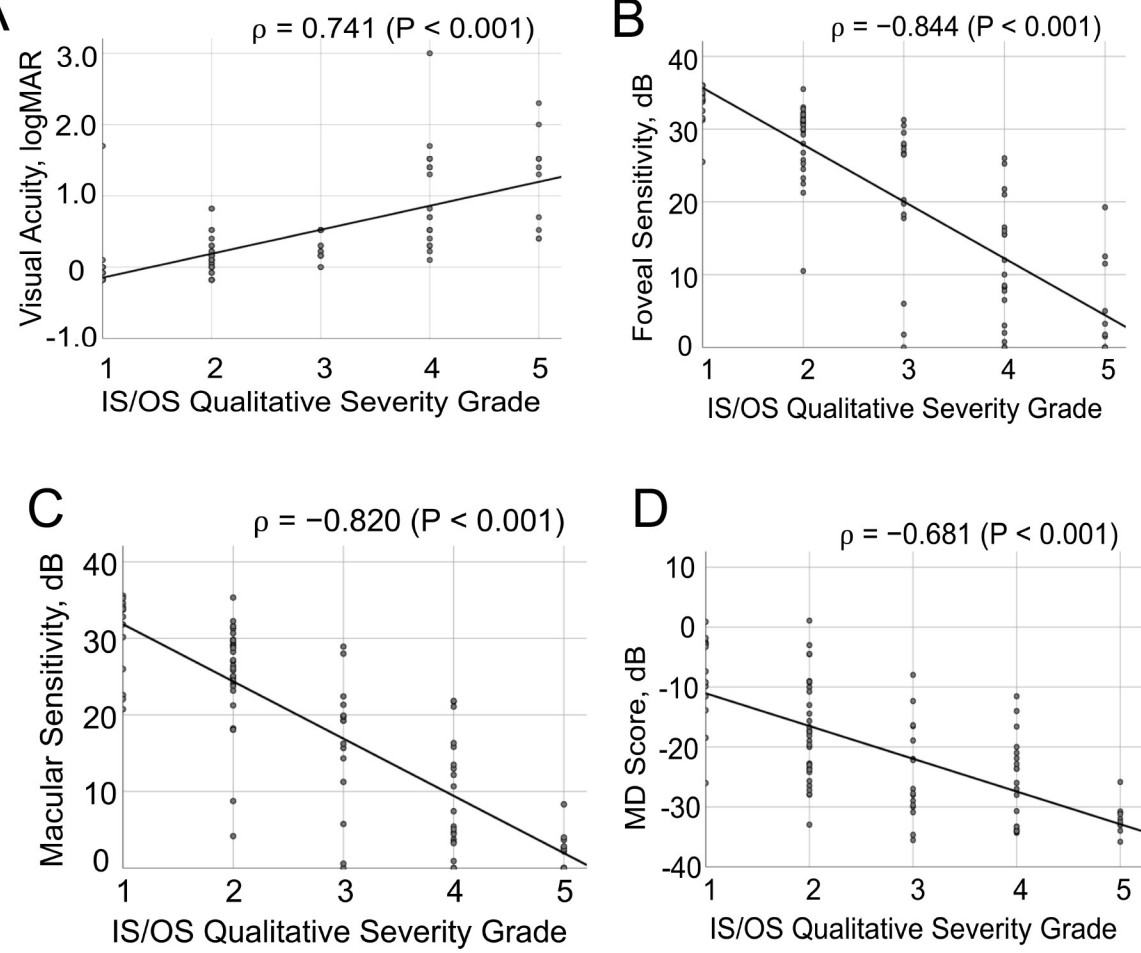

**Fig 5.** Plot diagrams showing correlations between the qualitative IS/OS severity grade and variables of the central retinal function, A, BCVA, B, foveal sensitivity (FS), C, macular sensitivity (MS) and D, mean deviation (MD) of the 10–2 program of the HFA.

## Discussion

In this study, we developed a qualitative grading system for the severity of photoreceptor IS/OS degeneration recorded in SD-OCT images and examined the validity of the IS/OS severity

**Table 3. Multiple regression analyses of parameters influencing central visual sensitivity.**

| | Model 1 | | | Model 2 | | | Model 3 | | |
|---|---|---|---|---|---|---|---|---|---|
| | | β | *P*-value | | β | *P*-value | | β | *P*-value |
| BCVA | IS/OS severity grade | 0.659 | <0.001 | IS/OS severity grade | 0.809 | <0.001 | IS/OS severity grade | 0.633 | <0.001 |
| | | | | CMT in 1mm | 0.247 | 0.012 | CMT in 1mm | 0.402 | <0.001 |
| | | | | | | | ONL thickness | −0.388 | 0.002 |
| FS | IS/OS severity grade | −0.820 | <0.001 | | | | | | |
| MS | IS/OS severity grade | −0.820 | <0.001 | | | | | | |
| MD | IS/OS severity grade | −0.674 | <0.001 | IS/OS severity grade | −0.388 | 0.001 | | | |
| | | | | IS-EZ width | 0.374 | 0.002 | | | |

β, standardized partial regression coefficient of multiple regression analyses.

grade in assessing the central retinal sensitivity of patients with RP. Although Sandberg et al. [1] and Aizawa et al. [7] have previously reported a qualitative grading system focusing on the morphologic changes of the IS/OS line (IS-EZ) using TD-OCT, their classifications were consisted of only three grades, and the resolution of TD-OCT is far below that of currently available SD-OCT. While there have been many reports regarding quantitative parameters, including the IS-EZ width, IS/OS thickness, ONL thickness and CMT in a 1-mm circle, that were found to be significantly correlated with the central visual sensitivity in patients with RP [1–24], this is the first report dealing with the qualitative evaluation of the IS/OS layer of SD-OCT images in patients with RP and statistically comparing it to other quantitative parameters. Although we confirmed that these quantitative parameters were significantly correlated with the central retinal sensitivity, our results further showed that the qualitative IS/OS severity grade was significantly highly correlated with the BCVA, FS and MS and revealed that the IS/OS severity grade the most highly influenced variables of the central retinal sensitivity, including BCVA, FS, MS and MD. Since Sandberg et al. and Aizawa et al. have previously reported that their qualitative severity grades using TD-OCT correlated with the BCVA [1, 7], it can be said that our current results using SD-OCT confirmed their results.

We contrived the current qualitative grading system based on the findings of previous SD-OCT studies using several animal models for human RP harboring known genetic mutations [25–31]. The objectives of these studies included identifying the background histologic and ultrastructural characteristics behind certain abnormal features of SD-OCT. Although SD-OCT noninvasively reveals fine structural changes in the retina, the precise pathological changes corresponding to certain abnormal findings of SD-OCT have been unclear, as it is impossible to perform histologic analyses of patients with RP. The information obtained from previous animal studies have been useful for speculating on the histologic and ultrastructural changes in human RP patients. The SD-OCT findings of these animal models have shown that the diffuse hyperreflective changes in the IS/OS layer and the thinning of the ONL layer are common abnormal findings of photoreceptor degeneration, regardless of the genetic mutations [25, 31]. The diffuse hyperreflective changes can be attributed to the phenomenon of broadening of the IS-EZ and IZ, resulting in their merging into a single hyperreflective band occupied in the IS/OS layer [26–28]. Although Joe et al. reported that the merging of the IS-EZ and IZ (like hyperreflective changes in the IS/OS layer) appeared to be an initial sign of retinal degeneration before the appearance of obvious structural and functional abnormalities [31], diffuse hyperreflective changes were observed even in the later stages of retinal degeneration where the photoreceptor outer segments were severely disorganized [25, 26, 28–30]. To summarize previous animal experiments, not only partial disarrangement of photoreceptor discs but also severely disorganized photoreceptor inner and outer segments lead to diffuse hyperreflective changes in the IS/OS layer in SD-OCT. Normally, the IS/OS layer is packed with regularly arranged IS and OS. This compactness may create optical uniformity and result in the hyporeflectivity in the space between IS-EZ and IZ in SD-OCT. We speculate that when this compactness is disordered by even mild pathologic changes in the IS/OS layer, the optical uniformity is lost and, consequently, the IS/OS layer becomes diffusely hyperreflective on SD-OCT [25–31]. We, therefore, concluded that the hyperreflective changes in the IS/OS layer or broad IS-EZ and/or IZ changes were a nonspecific sign of photoreceptor IS/OS degeneration, and covered a wide variety of pathologic changes, from mild to severe disorganization of the photoreceptor IS/OS, regardless of the causative gene mutations [25–31]. For these reasons, we selected hyperreflective changes in the IS/OS layer as a qualitative marker of photoreceptor IS/OS degeneration.

In the present study, the distribution of patient's severity appeared to slightly vary in grades 2–4 (Figs 4 and 5). We considered these phenotypic variations to be due to the phenomenon

of hyperreflective changes in the IS/OS layer comprising a wide variety of morphologic changes due to photoreceptor degeneration [25–31].

Our qualitative IS/OS severity grading system is simple and easy to use and may be useful for evaluating, to some extent, the retinal sensitivity in the central area in patients with RP. While static perimetry is necessary for assessing the patients' exact visual function, the findings may be influenced by the patients' general condition, their ability to concentrate and their level of understandings as it is basically a psychophysical examination. Conversely, because SD-OCT is an objective test, it is scarcely influenced by such subjective factors. Although SD-OCT cannot replace static perimetry, it may be a useful complementary test for assessing not only the structural but also the functional aspects of the retina in patients with RP.

Several limitations associated with the present study warrant mention. First, because this study was a retrospective cross-sectional study, the progression of RP was not considered. Future study should explore whether or not SD-OCT can reflect the progressive velocity of RP. Second, we did not consider patients' genetic heterogeneity. There is some variation in the overall severity among inheritance patterns and causative genes [33]. However, we did not perform a genetic study of the participants in the present study. Because most participants (86%) seemed to be simplex RP based on their family histories, we were unable to differentiate their exact inheritance patterns. Third, the reproducibility of the results of the visual field test may not be completely assured. Because the visual field test is basically a psychophysical examination, it is greatly influenced by patient's cooperation. To minimize this issue, we selected patients who underwent HFAs regularly in our clinic and were expected to be used to the assessment. Fourth, the validity of the current qualitative grading system should be further examined. Although the IS/OS severity grade was shown to most strongly influence the central retinal sensitivity among parameters of SD-OCT in this study, there may be a more suitable qualitative severity grade system. In addition, the proper combinations of qualitative and quantitative parameters may more accurately indicate patients' retinal sensitivity than individual parameters. Further studies will be needed to clarify these points.

In conclusion, the qualitative IS/OS severity grade system that we proposed in this report is simple and easy to perform and was most significantly correlated with the central retinal sensitivity among quantitative parameters of SD-OCT images that we examined. The qualitative IS/OS severity grade can be potentially applied to patients with wider variety of severity than the previously reported quantitative parameters covered. While patients' actual severity can vary within the same grade assignment, this severity grade can be used as a complementary evaluation of the central retinal sensitivity in patients with RP.

## Supporting information

**S1 Fig. Plot diagrams showing correlations between the SD-OCT parameters, qualitative IS/OS severity grade, IS-EZ width and IS/OS layer thickness, and variables of the central retinal function, BCVA, foveal sensitivity (FS), macular sensitivity (MS) and mean deviation (MD) of the 10–2 program of the HFA.**
(TIF)

**S2 Fig. Plot diagrams showing correlations between SD-OCT parameters, ONL thickness, central macular thickness (CMT) in 1.0- and 3.0-mm circles, and variables of the central retinal function, BCVA, FS, MS and MD of the 10–2 program of the HFA.**
(TIF)

**S1 Table. Raw data for the quantitative parameters of SD-OCT findings.**
(PDF)

**S2 Table. Raw data for the central retinal sensitivity.**
(PDF)

**S3 Table. Raw data for evaluation of the qualitative IS-OS severity grade by the two graders and final agreement.**
(PDF)

## Author Contributions

**Data curation:** Masaaki Saito, Yukihiko Suzuki.

**Formal analysis:** Aiko Hara, Mitsuru Nakazawa.

**Funding acquisition:** Mitsuru Nakazawa, Masaaki Saito, Yukihiko Suzuki.

**Investigation:** Aiko Hara, Mitsuru Nakazawa.

**Methodology:** Aiko Hara, Mitsuru Nakazawa.

**Project administration:** Mitsuru Nakazawa.

**Software:** Mitsuru Nakazawa.

**Supervision:** Mitsuru Nakazawa, Masaaki Saito, Yukihiko Suzuki.

**Validation:** Masaaki Saito, Yukihiko Suzuki.

**Writing – original draft:** Aiko Hara.

**Writing – review & editing:** Mitsuru Nakazawa, Masaaki Saito, Yukihiko Suzuki.

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
