## [Decision Letter · Decision Letter 0]

3 Mar 2020

PONE-D-20-03802

The Qualitative Assessment of Optical Coherence Tomography and the Central Retinal Sensitivity in Patients with Retinitis Pigmentosa

PLOS ONE

Dear Professor Nakazawa,

Thank you for submitting your manuscript to PLOS ONE. After careful consideration, we feel that it has merit but does not fully meet PLOS ONE’s publication criteria as it currently stands. Therefore, we invite you to submit a revised version of the manuscript that addresses the points raised during the review process.

Please address the concerns of the reviewers as stated below. In particular, please compare and contrast your current grading system to previously published results.  In addition, concern was raised about your statistical analysis by both reviewers, and reviewer two would like information about the histological correlates of hyperreflective changes in the EZ line.

We would appreciate receiving your revised manuscript by Apr 17 2020 11:59PM. To enhance the reproducibility of your results, we recommend that if applicable you deposit your laboratory protocols in protocols.io, where a protocol can be assigned its own identifier (DOI) such that it can be cited independently in the future. For instructions see: http://journals.plos.org/plosone/s/submission-guidelines#loc-laboratory-protocols

We look forward to receiving your revised manuscript.

Kind regards,

Alfred S Lewin, Ph.D.

Academic Editor

PLOS ONE

Journal Requirements:

'This study was supported, in part, by the Grant-in-Aids for Scientific Research 19K09926 (MN) and 16K11257 (MS) from the Japan Society for the Promotion of Science (Tokyo, Japan: https://www.jsps.go.jp/) and Research grants from Alcon (MN; Tokyo Japan), Novartis (MN; Tokyo, Japan), Santen (MN; Osaka, Japan), AMO (MN; Tokyo, Japan), HOYA (MN; Tokyo, Japan) and Pfizer (MN; Tokyo, Japan). The funders had no role in study design, data collection and analysis, decision to publish, or preparation of the manuscript.'

We note that you received funding from commercial sources: Alcon, Novartis, Santen, AMO and Pfizer.

a. Please provide an amended Competing Interests Statement that explicitly states these commercial funders, along with any other relevant declarations relating to employment, consultancy, patents, products in development, marketed products, etc.

Reviewers' comments:

Reviewer's Responses to Questions

**Comments to the Author**

1. Is the manuscript technically sound, and do the data support the conclusions?

Reviewer #1: Yes

Reviewer #2: Partly

2. Has the statistical analysis been performed appropriately and rigorously? 

Reviewer #1: Yes

Reviewer #2: No

3. Have the authors made all data underlying the findings in their manuscript fully available?

Reviewer #1: Yes

Reviewer #2: Yes

4. Is the manuscript presented in an intelligible fashion and written in standard English?

Reviewer #1: Yes

Reviewer #2: Yes

5. Review Comments to the Author

Reviewer #1: In this manuscript, Hara et al. present a paper analyzing OCT findings and retinal sensitivity as measured by HVF in 93 patients with RP. While the study of retinal function and OCTs has been previously described in the literature, the authors present a qualitative assessment of SD-OCT as categorized into 5 stages. The statistical analysis is sound and the conclusions are appropriately drawn.

1. In table 2, the authors present a negative Spearman correlation coefficient between foveal sensitivity and CMT at 1mm. This would imply that increases in central macular thickness would correlate with decreases in foveal sensitivity. This is not consistent with the correlations found in CMT at 3mm. The supplemental data suggests this correlation is positive. If it is indeed negative, a justification or explanation should be provided.

2. The authors describe a 5-stage categorization of SD-OCTs in this manuscript that has similarities to the three-stage grading system described in 2009 by Aizawa et al. A comparison to this previously described study should be included in the discussion section.

Overall considerations

Hara et al. have prepared a manuscript describing a set of OCT thickness and their correlation with measures of retinal sensitivity as measured by HVF. The authors suggest a qualitative staging system for SD-OCTs in patients with RP, however this study does have some similarities to the study by Aizawa et al. in 2009. Similarities and differences between the two should be discussed and afterwards this manuscript may be considered as a validation study of prior reports.

Reviewer #2: The authors defined a new severity criteria to classify the OCT findings in RP patients, and showed that the grades are highly correlated with central visual status. This is an interesting study, but there are several points that should be addressed to improve the study.

1. The authors defined the OCT severity grade based on the their previous laboratory study to compare the OCT image and histological findings. They focused on hyper-reflective changes of EZ in OCT of RP patients (i.e. grade 2 and 3); however, the histological findings that correspond to EZ hyper-reflectance is unclear from the description of the current Ms. In Page 12, the authors described that “the hyperreflective changes in the IS/OS layer or … were a nonspecific sign of photoreceptor IS/OS degeneration”. It is unclear what is “nonspecific sign”. I would like to know what histological findings (Light or electron microscopy) correspond to the hyper-reflective changes of EZ.

2. They showed the representative severity grade of OCT findings in Fig. 2. However, it is difficult to know how they define the “hyper-reflective EZ”. Any internal control (e.g. the reflectivity of ILM/ELM/RPE or others) to determine the “hyper-reflectance” of EZ? The differences between partial (grade 2) vs. diffuse (grade 3) hyper-reflective changes are also unclear. The authors should provide more objective and concrete methods that can be repeated by independent researchers/clinicians.

3. The relationships between severity grade in OCT (grade 1-5) and visual function (VA, FS, MS, and MD) were assessed by Spearman’s correlation rank test. It would be reasonable to analyze continuous variables (e.g. retinal sensitivity, EZ lengths, retinal thickness, etc.) in Spearman’s correlation test, but applying this statistics to non-continuous variables (i.e. grade 1-5) is not appropriate.

4. The work imply that qualitative OCT severity grade is more strongly correlated with macular function rather than qualitative parameters (i.e. EZ width, IS/OS thickness, ONL thickness, CMT, etc.). However, as described above, qualitative and quantitative values cannot be directly compared. In addition, in clinical settings, quantitative, rather than qualitative, values are usually considered to be more objective and deemed appropriate for outcome measures. I wonder what is the rationale to propose ambiguous grading system rather than using objective and continuous measurements such as EZ length and ONL thickness. Please justify the rationale to use subjective grading.

5. Fig. 3. Thickness of photoreceptor IS/OS layer were indicated the arrow 2. However, the bar seems to indicate the length between ELM (but not EZ) to RPE. The method how they define IS/OS thickness should be described.

6. PLOS authors have the option to publish the peer review history of their article (what does this mean?). If published, this will include your full peer review and any attached files.

Reviewer #1: No

Reviewer #2: No

---

## [Author Response · Author response to Decision Letter 0]

3 Apr 2020

Responses to the reviewers’ comments

Reviewer #1

1. In table 2, the authors present a negative Spearman correlation coefficient between foveal sensitivity and CMT at 1mm. This would imply that increases in central macular thickness would correlate with decreases in foveal sensitivity. This is not consistent with the correlations found in CMT at 3mm. The supplemental data suggests this correlation is positive. If it is indeed negative, a justification or explanation should be provided.

Response

It was our mistake and the correlation coefficient of this particular portion was indeed a positive correlation. I have corrected Table 2 in the revised version of the manuscript.

2. The authors describe a 5-stage categorization of SD-OCTs in this manuscript that has similarities to the three-stage grading system described in 2009 by Aizawa et al. A comparison to this previously described study should be included in the discussion section.

Responses

As pointed out by the reviewer, Aizawa et al. [7] and Sandberg et al. [1] have previously discussed the qualitative severity grades using time-domain OCT (TD-OCT). We have included their previous reports and further discussed the difference between the findings obtained from their previous studies and those from our current analysis using SD-OCT with far better visibility than TD-OCT.

To clarify these points, we have added sentences “In addition, while Sandberg et al. and Aizawa et al. have previously reported the correlation between the qualitative changes in time-domain OCT (TD-OCT) findings and central visual acuity in patients with RP [1, 7], the resolution of TD-OCT was limited compared with that of SD-OCT.” in L88-91 in P4, “We considered these characteristic changes in the IS/OS layer on SD-OCT observed in animal models of RP to be applicable for classifying the severity of morphologic changes in patients with RP in greater detail than those previously reported [1, 7]. ” in L107-110 in P5, “Although Sandberg et al. [1] and Aizawa et al. [7] have previously reported a qualitative grading system focusing on the morphologic changes of the IS/OS line (IS-EZ) using TD-OCT, their classifications were consisted of only three grades, and the resolution of TD-OCT is far below that of currently available SD-OCT.” in L279-283 in P12, and “Since Sandberg et al. and Aizawa et al. have previously reported that their qualitative severity grades using TD-OCT correlated with the BCVA [1, 7], it can be said that our current results using SD-OCT confirmed their results.” in L292-294, in P13 of the revised manuscript.

Overall considerations

Hara et al. have prepared a manuscript describing a set of OCT thickness and their correlation with measures of retinal sensitivity as measured by HVF. The authors suggest a qualitative staging system for SD-OCTs in patients with RP, however this study does have some similarities to the study by Aizawa et al. in 2009. Similarities and differences between the two should be discussed and afterwards this manuscript may be considered as a validation study of prior reports.

Responses

As mentioned in the above section, we have discussed the similarities and differences between the previous studies reported Aizawa et al. and Sandberg et al. [1, 7] in the Introduction and Discussion sections in L88-91 in P4, L107-110 in P5, L279-283 in P12, and L292-294, in P13 of the revised manuscript. We have added references 7 and 8 in the revised manuscript.

Reviewer #2

1. The authors defined the OCT severity grade based on their previous laboratory study to compare the OCT image and histological findings. They focused on hyper-reflective changes of EZ in OCT of RP patients (i.e. grade 2 and 3); however, the histological findings that correspond to EZ hyper-reflectance is unclear from the description of the current Ms. In Page 12, the authors described that “the hyperreflective changes in the IS/OS layer or … were a nonspecific sign of photoreceptor IS/OS degeneration”. It is unclear what is “nonspecific sign”. I would like to know what histological findings (Light or electron microscopy) correspond to the hyper-reflective changes of EZ.

Responses

To more clearly explain the light and electron microscopic findings of the photoreceptor IS/OS degeneration in the animal experiments, we have added the following sentences to the revised manuscript, “As degeneration of the photoreceptor progresses, although the IS-EZ and interdigitation zone (IZ) are initially identified as two independent discrete lines, they gradually broaden and finally merge into a single diffuse hyperreflective zone in the IS/OS layer [26-28]. The diffuse hyperreflective changes in the IS/OS layer are considered to be the same phenomenon as merging of the IS-EZ and IZ [31]. We considered these characteristic changes in the IS/OS layer on SD-OCT observed in animal models of RP to be applicable for classifying the severity of morphologic changes in patients with RP in greater detail than those previously reported [1, 7].” in L103-110 in P5 in the Introduction section, “and “The diffuse hyperreflective changes can be attributed to the phenomenon of broadening of the IS-EZ and IZ, resulting in their merging into a single hyperreflective band occupied in the IS/OS layer [26-28].” In L306-308 in P13, and “To summarize previous animal experiments, not only partial disarrangement of photoreceptor discs but also severely disorganized photoreceptor inner and outer segments lead to diffuse hyperreflective changes in the IS/OS layer in SD-OCT. Normally, the IS/OS layer is packed with regularly arranged IS and OS. This compactness may create optical uniformity and result in the hyporeflectivity in the space between IS-EZ and IZ in SD-OCT. We speculate that when this compactness is disordered by even mild pathologic changes in the IS/OS layer, the optical uniformity is lost and, consequently, the IS/OS layer becomes diffusely hyperreflective on SD-OCT [25-31].” in L312-320 in P13-14 in the Discussion section.

Regarding the words “nonspecific sign”, to further explain the meaning, we have changed the sentences into “We, therefore, concluded that the hyperreflective changes in the IS/OS layer or broad IS-EZ and/or IZ changes were a nonspecific sign of photoreceptor IS/OS degeneration, and covered a wide variety of pathologic changes, from mild to severe disorganization of the photoreceptor IS/OS, regardless of the causative gene mutations [25-31].” in L320-323 in P14.

2. They showed the representative severity grade of OCT findings in Fig. 2. However, it is difficult to know how they define the “hyper-reflective EZ”. Any internal control (e.g. the reflectivity of ILM/ELM/RPE or others) to determine the “hyper-reflectance” of EZ? The differences between partial (grade 2) vs. diffuse (grade 3) hyper-reflective changes are also unclear. The authors should provide more objective and concrete methods that can be repeated by independent researchers/clinicians.

Responses

We have added the introduction of background research findings regarding the formation of the diffuse hyperreflective change in the IS/OS layer as “As degeneration of the photoreceptor progresses, although the IS-EZ and interdigitation zone (IZ) are initially identified as two independent discrete lines, they gradually broaden and finally merge into a single diffuse hyperreflective zone in the IS/OS layer [26-28].” in L103-105, and “We considered these characteristic changes in the IS/OS layer on SD-OCT observed in animal models of RP to be applicable for classifying the severity of morphologic changes in patients with RP in greater detail than those previously reported [1, 7].” in L107-110 in P5 in the Introduction section, as mentioned above.

In addition, to clarify the difference between grades 2 and 3, we have changed the definition of grades 1-3 as “The definition of these five grades were as follows: grade 1, almost normal foveal structure or a sharply defined IS-EZ and IZ at the foveal center; grade 2, partial hyperreflective change in the IS/OS layer of the foveal region or a broadened IZ and/or IS-EZ without merging; grade 3, diffuse hyperreflective changes in the IS/OS layer or broad IS-EZ changes in the foveal region or merging of the broadened IS-EZ and IZ” in L165-170 in P7. We have also added yellow arrows in Fig 2 to help readers easily identify the findings, and changed the legend of Fig 2 as “Sample SD-OCT pictures indicating each severity grade. Definition of grades: grade 1, almost a normal foveal structure or sharply defined IS-EZ and IZ at the foveal center (yellow arrow); grade 2, partial hyperreflective changes in the IS/OS layer of the foveal region or a broadened IZ (yellow arrow) and/or IS-EZ without merging; grade 3, diffuse hyperreflective changes in the IS/OS layer or broad IS-EZ changes in the foveal region or merging of the broadened IS-EZ and IZ (yellow arrow); grade 4, remnants of the IS/OS-like structure on the retinal pigment epithelium (yellow arrow); grade 5, absent IS/OS-like structure (yellow arrow).” in L469-475 in P19, to help the putative readers understand more easily. 

3. The relationships between severity grade in OCT (grade 1-5) and visual function (VA, FS, MS, and MD) were assessed by Spearman’s correlation rank test. It would be reasonable to analyze continuous variables (e.g. retinal sensitivity, EZ lengths, retinal thickness, etc.) in Spearman’s correlation test, but applying this statistics to non-continuous variables (i.e. grade 1-5) is not appropriate.

Responses

We have added the box plots as a new Fig 4 to show the relationship between the variables reflecting visual function (BCVA, FS, MS and MD) and the severity grade in SD-OCT. And we also performed ANOVA using a Kruskal-Wallis test following a Bonferroni’s test as a post hoc multiple comparison analysis to compare the median value in each score group. Since Fig 4 A-D indicated that these variables appeared to be well correlated with the severity score, we considered that our 5-grade severity score can be treatable as a semi-interval scale and applied a Spearman’s correlation rank test. To clarify these points, we have added the sentences “Values of variables reflecting the central retinal sensitivity in each qualitative severity grade group were compared by a one-way analysis of variance using the Kruskal-Wallis test, followed by Bonferroni’s test as a post hoc multiple comparison analysis.” in L203-206 in P8, “While, strictly speaking, the qualitative severity grade score is not an interval scale but an ordinal scale, we considered that the qualitative severity grade score to be treatable as a semi-interval scale followed by assessment by the Spearman’s correlation rank test.” in K210-212 in P8, and “The median values of the BCVA, FS, MS or MD in each qualitative IS/OS severity group were significantly different (P < 0.001, Kruskal-Wallis test). The relationships between values reflecting the central retinal function and the qualitative IS/OS severity grades are presented in Fig 4. Regarding the BCVA, the median value in grades 3 was significantly better than that in grade 4 (P < 0.001, Bonferroni’s test, Fig 4). Regarding the FS and MS, the median values in grades 2 and 3 were significantly greater than those in grades 3 and 4, respectively (P = 0.003, FS, grade 2 vs. grade 3; P < 0.001, FS, grade 3 vs. grade 4, MS, grade 2 vs. grade 3, grade 3 vs. grade 4, Bonferroni’s test, Fig 4). Regarding the MD, the median values in grades 1 and 2 were significantly greater than those in grades 2 and 3, respectively (P = 0.004, grade 1 vs. grade 2; P = 0.032, grade 2 vs. grade 3, Bonferroni’s test, Fig 4). Given the results shown in Fig 4 apparently reflecting the central retinal function were correlated with the qualitative IS/OS severity grade, we considered it reasonable that the qualitative IS/OS severity grade was considered as a semi-quantitative value.” in L230-242 in P9-10 in the Results section.

4. The work imply that qualitative OCT severity grade is more strongly correlated with macular function rather than qualitative parameters (i.e. EZ width, IS/OS thickness, ONL thickness, CMT, etc.). However, as described above, qualitative and quantitative values cannot be directly compared. In addition, in clinical settings, quantitative, rather than qualitative, values are usually considered to be more objective and deemed appropriate for outcome measures. I wonder what is the rationale to propose ambiguous grading system rather than using objective and continuous measurements such as EZ length and ONL thickness. Please justify the rationale to use subjective grading.

Responses

As mentioned in the above section, the newly added box plot in Fig 4 indicated that these variables appeared to be well correlated with the severity score, we considered that our 5-grade severity score can be treatable as a semi-interval scale and applied a multiple regression analysis. We tried the validity of the qualitative severity grade by comparing to other quantitative parameters by a common test. To clarify this point, we have added a sentence “In this analysis, we also considered the qualitative severity grade score as a semi-interval scale and applied a multiple regression analysis as well.” in L215-216 in P8.

As was pointed out by the reviewer, quantitative parameters are usually considered to be more objective and reasonable to handle for a comparison study in the clinical settings. However, in the previously reported quantitative analyses, the subjects were restricted only to patients with visual acuity better than a certain level or measurable structural changes. Consequently, they excluded patients with poor visual acuity or those without measurable IS/EZ width and/or IS/OS thickness on OCT. The rationale to propose the current qualitative severity grade is that this scoring system can basically be applied to patients with RP having any kind of severity and can be easily performed by any clinical ophthalmologists.

To clarify these points, we have added a sentence “However, in these quantitative analyses, the subjects should be restricted to patients with visual acuity better than a certain level or measurable structural changes for analyzing the correlation between structural parameters and the visual function [1, 9, 10, 12, 13, 15, 18, 20, 21].” in L85-88 in P4, and a sentence “The qualitative IS/OS severity grade can be potentially applied to patients with wider variety of severity than the previously reported quantitative parameters covered.” in L357-359 in P15.

In addition, the reason why our severity grading system is simple and easy (see L356-360 in P15) is that we can just take a look at and focus on the SD-OCT findings in the central foveal portion. To indicate this point, we have added yellow arrows in Fig 2.

5. Fig. 3. Thickness of photoreceptor IS/OS layer were indicated the arrow 2. However, the bar seems to indicate the length between ELM (but not EZ) to RPE. The method how they define IS/OS thickness should be described.

Responses

I understand that the photoreceptor inner segment consists of the myoid at the inner portion and the ellipsoid at the outer portion and that the myoid area locates adjacent to the external limiting membrane (ELM). Since the distal tip of the photoreceptor outer segment attaches the inner surface of the RPE, the thickness of the photoreceptor inner and outer segment (IS/OS) layer is defined as the distance between ELM and the inner surface of the RPE. According to this definition, we measured the length between ELM and RPE to obtain the thickness of the IS/OS layer. If the distance between EZ and RPE was measured, the length of the inner segment myoid area was missed and consequently the obtained value would not represent the IS/OS thickness. I believe that the arrow 2 in Fig 3 is correct and it does not need to be changed.

To clarify this point, we have added a phrase “as the photoreceptor inner and outer segments are normally located in the space between ELM and RPE.” in L187-188 in P7.

---

## [Decision Letter · Decision Letter 1]

21 Apr 2020

The Qualitative Assessment of Optical Coherence Tomography and the Central Retinal Sensitivity in Patients with Retinitis Pigmentosa

PONE-D-20-03802R1

Dear Dr. Nakazawa,

We are pleased to inform you that your manuscript has been judged scientifically suitable for publication and will be formally accepted for publication once it complies with all outstanding technical requirements.

With kind regards,

Alfred S Lewin, Ph.D.

Section Editor

PLOS ONE

Additional Editor Comments (optional):

Reviewers' comments:

Reviewer's Responses to Questions

**Comments to the Author**

1. If the authors have adequately addressed your comments raised in a previous round of review and you feel that this manuscript is now acceptable for publication, you may indicate that here to bypass the “Comments to the Author” section, enter your conflict of interest statement in the “Confidential to Editor” section, and submit your "Accept" recommendation.

Reviewer #2: All comments have been addressed

2. Is the manuscript technically sound, and do the data support the conclusions?

Reviewer #2: (No Response)

3. Has the statistical analysis been performed appropriately and rigorously? 

Reviewer #2: (No Response)

4. Have the authors made all data underlying the findings in their manuscript fully available?

Reviewer #2: (No Response)

5. Is the manuscript presented in an intelligible fashion and written in standard English?

Reviewer #2: (No Response)

6. Review Comments to the Author

Reviewer #2: (No Response)

7. PLOS authors have the option to publish the peer review history of their article (what does this mean?). If published, this will include your full peer review and any attached files.

Reviewer #2: No

---

## [Editor Report · Acceptance letter]

23 Apr 2020

PONE-D-20-03802R1 

The Qualitative Assessment of Optical Coherence Tomography and the Central Retinal Sensitivity in Patients with Retinitis Pigmentosa 

Dear Dr. Nakazawa:

I am pleased to inform you that your manuscript has been deemed suitable for publication in PLOS ONE. Congratulations! Your manuscript is now with our production department. 

With kind regards,

on behalf of

Dr. Alfred S Lewin 

Section Editor

PLOS ONE